# Genotype–Phenotype Correlation of EVC Variants in Ellis-Van Creveld Syndrome: A Systematic Review and Case Report

Sandra Rodriguez-Cambranis [1,2], Addy-Manuela Castillo-Espinola [3], Claudia-Daniela Fuentelzas-Rosado [4], Paulina Salazar-Sansores [4], Claudia-Gabriela Nuñez-Solis [4], Hugo-Antonio Laviada-Molina [4], Aurea-Karina Zetina-Solorzano [2] and Felix-Julian Campos-Garcia [4,5,6,*]

1    Pediatrics Residency Program, School of Medicine, Autonomous University of Yucatan, Merida 97000, Yucatan, Mexico
2    Pediatrics Department, General Hospital "Dr. Agustin O'Horan", Merida 97000, Yucatan, Mexico
3    Cardiology Pediatrics Department, IMSS UMAE, Merida 06720, Yucatan, Mexico; addy.castillo@imss.gob.mx.com
4    Research Department, Marista University, Merida 97302, Yucatan, Mexico; cfuentelzas1814049@a.marista.edu.mx (C.-D.F.-R.); psalazar1814116@a.marista.edu.mx (P.S.-S.); cnunez1711025@a.marista.edu.mx (C.-G.N.-S.); hlaviada@marista.edu.mx (H.-A.L.-M.)
5    Doctoral Program in Medical Sciences, National Autonomous University of Mexico, Mexico City 04510, Mexico
6    Genetics Department, General Hospital "Dr. Agustin O'Horan", Merida 97000, Yucatan, Mexico
*    Correspondence: felix.campos@ssy.gob.mx

**Abstract:** Ellis-van Creveld syndrome (EvC) is a rare genetic disorder (7:10,000,000) caused by biallelic pathogenic variants in *EVC* and *EVC2*, which are located in close proximity on chromosome 4p16.2 in a divergent orientation. These genes encode ciliary complex proteins essential for Hedgehog signaling. EvC is characterized by congenital heart disease (CHD), postaxial polydactyly, and rhizomelic shortening. We present a case of a female newborn from southeast Mexico carrying a novel missense variant in *EVC*, which is aligned with a systematic review aimed at exploring genotype–phenotype correlations in EVC-related EvC. A PRISMA-based systematic review was conducted in PubMed, Web of Science, and OVID/Medline (until December 2024). Studies reporting *EVC* variants in EvC were included. Data extraction and quality assessment were performed independently by four reviewers, and genotype–phenotype correlation analysis was conducted. Fifteen studies (*n* = 66 patients) met the inclusion criteria. The most prevalent features were postaxial polydactyly (95.5%), nail hypoplasia (68.2%), and CHD (66.7%) with atrioventricular canal as the most frequent subtype. Fifty-five distinct *EVC* variants across 132 alleles were identified, predominantly affecting the N-terminal region (first 699 amino acids). They were syndactyly correlated with pathogenic variants in exons 6, 12, and 13, which were proximal to the second and third coiled-coil domains. This review confirms the key clinical features of *EVC*-related EvC and highlights genetic heterogeneity. The correlation between syndactyly and specific exonic variants suggests potential genotype–phenotype associations, warranting further functional studies.

**Keywords:** Ellis-van Creveld syndrome; EVC; congenital heart disease

## 1. Introduction

Ellis-van Creveld (EvC) syndrome (OMIM: 225500) is a rare chondroectodermal dysplasia with an estimated prevalence of approximately 7 cases per 10 million individuals. However, epidemiological studies in specific populations report significantly higher prevalence rates, reaching up to 5 cases per 1000 live births [1] and as high as 1 in 60,000 live

births in populations with elevated rates of consanguinity [2]. EvC syndrome is classified as skeletal disorders caused by abnormalities of cilia or ciliary signaling in the latest nosology of genetic skeletal disorders [3]. This category includes 61 distinct diagnoses, six of which correspond to chondroectodermal dysplasia or EvC syndrome. The syndrome is primarily caused by biallelic pathogenic variants in *EVC* or *EVC2*. Additionally, other genes such as *WDR35*, *DYNC2LI1*, *GLI1*, and *SMO* have been implicated in related ciliopathies [4–6]. Several conditions previously associated with EvC syndrome are now classified under short-rib thoracic dysplasia, which is a group of skeletal dysplasias with overlapping clinical features [3]. Moreover, an increasing number of EvC-like phenotypes have been linked to heterozygous pathogenic variants in *PRKACA* and *PRKACB* [7] and homozygous pathogenic variants in *DYNC2H1* [8].

The *EVC* and *EVC2* genes are in close proximity on chromosome 4p16.2 and are arranged in a divergent orientation with their translational start sites separated by only 2.86 kb in the human genome [9]. This unique configuration suggests that *EVC* and *EVC2* may be co-regulated and functionally interdependent. Both genes encode transmembrane proteins that form a heterodimer at the base of the primary cilium, which is a specialized organelle crucial for Hedgehog (Hh) signaling. The EVC/EVC2 complex modulates the transduction of Hedgehog signals, which is a pathway essential for limb development, craniofacial morphogenesis, and endochondral ossification. Disruptions in this signaling cascade result in the characteristic skeletal and ectodermal defects observed in EvC syndrome [10].

The EVC protein consists of 992 amino acids and contains a single transmembrane helix spanning amino acids 24 to 46, which anchors it to the ciliary membrane. The remaining portion of the protein is cytoplasmic and features three coiled-coil domains: the first spanning amino acids 52–72, the second 262–282, and the third 696–724 [11]. These domains are crucial for EVC/EVC2 heterodimerization, stabilizing the protein complex at the ciliary membrane, and facilitating Hedgehog signal transduction. Pathogenic variants in *EVC* frequently disrupt these functional regions, impairing proper Hedgehog pathway regulation [12].

Clinically, EvC syndrome presents a classic triad of postaxial polydactyly, ectodermal dysplasia, and short-limb dwarfism [13,14]. Additional features include congenital heart disease (CHD), present in up to 66.7% of affected individuals, with atrioventricular canal defects being the most frequently reported cardiac anomaly [15]. Orofacial abnormalities such as nail dysplasia, dental anomalies, and cleft palate are also characteristic of the syndrome. However, significant variability exists in disease severity and phenotypic expression, underscoring the importance of genotype–phenotype correlation studies [16].

In this systematic review, we analyzed 66 patients from 49 distinct families, compiling phenotypic and genotypic data from 15 studies. We aimed to assess the correlation between variant location within the *EVC* gene and specific clinical manifestations, providing insights into the molecular basis of disease heterogeneity.

## 2. Materials and Methods

### 2.1. Clinical Case Description

We present the case of a female newborn from southeast Mexico with a novel missense variant in the *EVC* gene. Informed consent was obtained from both parents for the use of clinical and sociodemographic data as well as for the inclusion of images to support the case report. Furthermore, this study was reviewed and approved by the Ethics and Research Committee of General Hospital "Dr. Agustín O'Horan" under registration number CI-015-1-2025, and the approval date is 13 March 2025.

*2.2. Clinical Case: Genetic Analysis*

A peripheral blood sample was collected and stored as a dried blood spot (DBS) for whole exome sequencing analysis. Genomic DNA was extracted from DBS specimens using standard protocol. Exome capture was performed using xGen Exome Research Panel v2, which was supplemented with xGen human mtDNA and xGen Custom Hyb Panel v1 (Integrated DNA Technologies, Coralville, IA, USA). Sequencing was carried out on the NovaSeq X platform (Illumina, San Diego, CA, USA) generating 10,841,106,122 bases and uniquely aligned to the Genome Reference Consortium Human Build 38 (GRCh38) using the BWA-MEM2 protocol within the Galaxy Project. Approximately 99.4% of the targeted bases were covered at a depth greater than 20x. Alignments were visualized and analyzed using IGV software (Ver. 2.15.4). Variant calling was conducted using the DeepVariant protocol and analyzed on the Franklin platform by Genoox.

*2.3. Search Strategy*

A literature search was conducted using systematic review procedures in accordance with PRISMA guidelines. PUBMED, OVID MEDLINE and Web of Science databases were searched from July 2024 to December 2024 using the combinations of the next Medical Subject Heading (MeSH) terms: «Ellis-Van-Creveld», «Ellis-Van-Creveld Syndrome», and «EVC» (Table 1). Only articles published between 2004 and 2024 were considered for inclusion. All identified abstracts were uploaded to Rayyan® (Cambridge, MA, USA) to delete duplicated articles and to filter articles based on their title and abstract [2,17–30].

**Table 1.** Search strategy conducted from July 2024 to December 2024 with the MeSH terms and keywords used in different databases.

| Database | MeSH Terms and Search Strings |
|---|---|
| Ovid MEDLINE | Ellis-van Creveld.mp. and EVC.ab,ti. [mp=title, book title, abstract, original title, name of substance word, subject heading word, floating sub-heading word, keyword heading word, organism supplementary concept word, protocol supplementary concept word, rare disease supplementary concept word, unique identifier, synonyms, population supplementary concept word, anatomy supplementary concept word] |
| PubMed | (Ellis-van Creveld syndrome [Title/Abstract]) AND (EVC[Title/Abstract]) |
| Web of Science | (((TI=(Ellis-van Creveld)) AND AB=(EVC) |

*2.4. Inclusion and Exclusion Criteria*

2.4.1. EVC Variant Overview

All articles were reviewed for the presence of *EVC* variants. Only original research articles presenting clinical case reports involving EVC were considered. In studies reporting multiple patients, only those with clearly identified genetic variants and corresponding clinical phenotypes were included. The variant pathogenicity according to ACMG criteria was also included.

2.4.2. EVC Patient Summary

Only patients with biallelic pathogenic variants were considered. Articles that did not reference *EVC*, did not involve individuals with Ellis-van Creveld diagnosis or attributed the Ellis-van Creveld syndrome to pathogenic variants in other genes were not included.

Furthermore, any types of studies that were not case reports or described *EVC* variants in non-human subjects were also excluded.

*2.5. Data Extraction*

The extracted data encompassed variables describing general information and phenotype. General variables included sex, parental consanguinity, and family history. Phenotype-related variables covered nucleotide and protein changes as well as clinical presentation. Clinical phenotype data were categorized into the following variables: congenital heart disease (based on the International Pediatric and Congenital Cardiac Code; IPCCC [31]), facial features (e.g., thin upper lip, short frenula, bifid tongue tip, absence of the upper mucobuccal ridge, serrated alveolar ridge, long philtrum), dental anomalies (e.g., neonatal teeth, central incisors, hypodontia), skeletal features (e.g., postaxial polydactyly, axial polydactyly, syndactyly, rhizomelic shortening), and radiological findings (e.g., short tubular bones, metaphyseal widening, short ribs with a narrow chest, scoliosis, small iliac bones with downward spikes, trident-shaped acetabular roof). All extracted variables were organized into an Excel spreadsheet for further analysis. If a specific variable was not mentioned in a study, it was left blank (Table S1).

*2.6. Outcome Variables*

The primary objectives of this study were to identify the most frequently reported clinical features of Ellis-van Creveld syndrome and to evaluate potential correlations between variant positions within the genomic sequence and specific clinical manifestations. Data analysis was conducted using R software (v.1.2.5042; R Software, Inc., University of Auckland, New Zealand); we systematically mapped all reported clinical findings and their corresponding genetic variants onto both the gene and the protein structures.

## 3. Results

*3.1. Case Description: Clinical Findings*

We present a 15-day-old female newborn of Mayan–Yucatecan ancestry with no known history of consanguinity. However, both parents are residents of a Mayan population, suggesting potential endogamy. The parents are healthy with no history of chronic diseases or relevant surgical interventions. The mother's pregnancy was confirmed in the first trimester with routine prenatal care showing no complications, and prenatal ultrasounds were normal until the third trimester when possible fetal growth restriction was noted. The proband was delivered via cesarean section at 39 weeks of gestation with an APGAR score of 8/9 and no signs of respiratory distress. Physical examination revealed a weight of 2960 g and a length 48 cm ($-1.2$ SD). The propositus had a normocephalic head (head circumference: 34 cm, $-0.2$ SD), a broad forehead, horizontal palpebral fissures, neonatal teeth, a short neck, and a narrow thorax. Limb examination showed rhizomesomelic shortening, bilateral postaxial polydactyly in both hands, and axial polydactyly in the right foot (duplication of the 4th toe) with syndactyly of 4th and 5th toes (Figure 1A,B). Nail dysplasia was also observed. External genitalia were phenotypically female with no abnormalities.

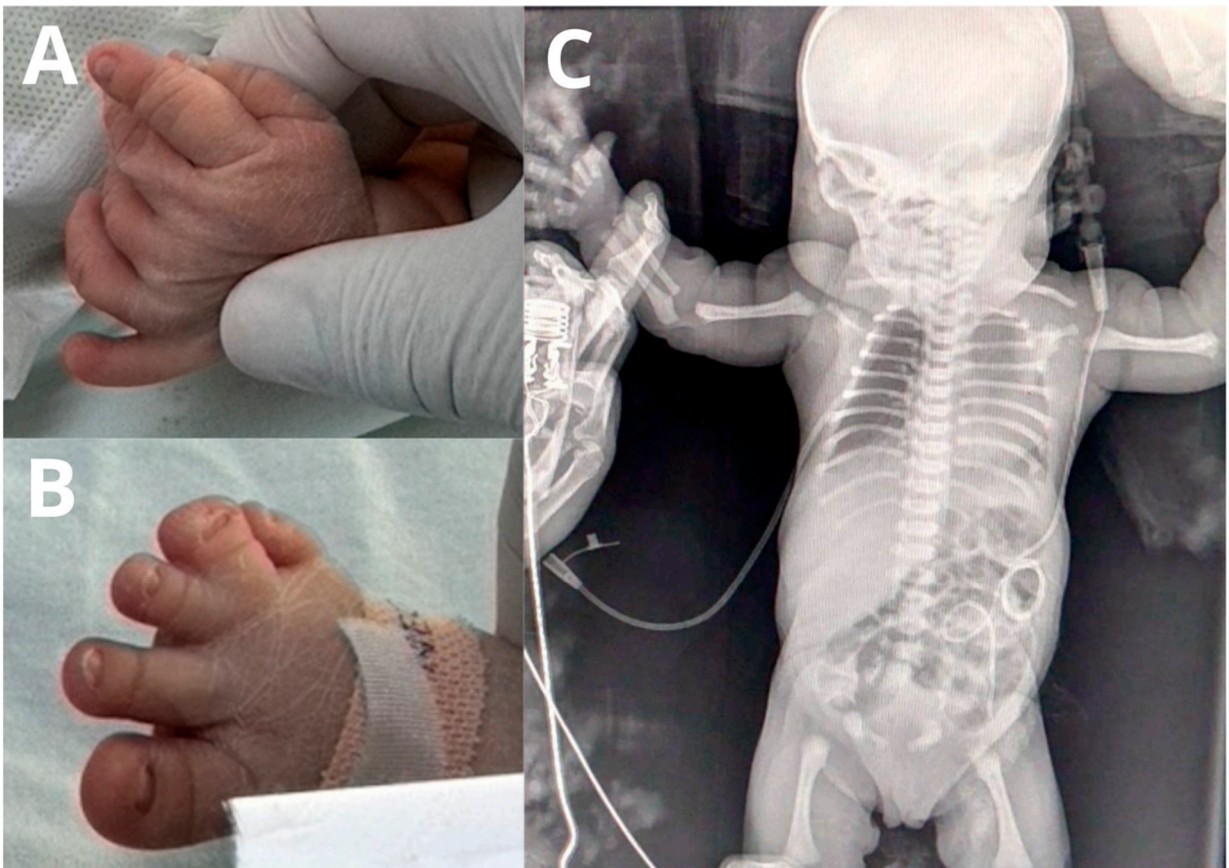

**Figure 1.** Clinical and radiological phenotype of propositus. (**A**) Left hand showing postaxial polydactyly. (**B**) Right foot showcasing syndactyly of the fourth and fifth toe along with axial polydactyly due to duplication of the fourth toe. (**C**) X-rays portraying a narrow thorax with short ribs and rhizomesomelic shortening of the limbs. Shortening of the pubic bones as well as a small iliac bones with a downward spike.

Short after the delivery, the patient was admitted to the Neonatal Intensive Care Unit (NICU) due to respiratory distress, which required continuous positive airway pressure (CPAP) and nutritional support with parenteral nutrition. After 4 days, the respiratory distress ceased and the patient was admitted to the Pediatrics department, where she received supplemental oxygen therapy and initiated oral feeding. Due to skeletal abnormalities and bone dysplasia, a referral was made to the Genetics department. Following clinical evaluation, imaging and diagnostic studies were requested, including whole exome sequencing, which confirmed the diagnosis of Ellis-van Creveld syndrome.

X-rays revealed shortened ribs and iliac bones in the cephalocaudal dimension. The ilium displays a downward-directed, hook-like projection at the greater sciatic notch. The ischial and pubic bones were also shortened along with notable shortening of the humerus, radius and ulna, which was accompanied by metaphyseal widening (Figure 1C).

The echocardiogram revealed situs solitus with levocardia, pulmonary veins draining into a single atrium, a single atrioventricular valve with insufficiency, two well-differentiated ventricles, and a small anterior malalignment ventricular septal defect (2 mm). Additionally, aortic hypoplasia was observed throughout the tract including emergence, proximal, and distal arch. Diuretic therapy was initiated with spironolactone (0.5–1 mg/kg/day), which was discontinued after a few days due to hyperkalemia and subsequently replaced with hydrochlorothiazide (1.0 mg/kg/day). The patient suffered sudden death at 20 days of life due to complex congenital heart disease.

### 3.2. Case Description: Results and Interpretation of Whole-Exome Sequencing

WES identified a likely pathogenic variant at locus Chr4-5793613G>A, EVC(NM_153717.3):c.1782G>A(p.Trp594*) in homozygosity. This variant involves a guanine-to-adenine substitution at nucleotide position 1782 in exon 13 of the *EVC* gene, leading to a nonsense variant that introduces a premature stop codon at position 594, truncating the protein. According to ACMG guidelines, this variant is classified as likely pathogenic based on the following criteria: null variant in a gene where loss of function is a known mechanism of disease (PVS1) and an extremely low frequency in gnomAD population databases (PM2).

### 3.3. Publications and EVC Variants

Four independent researchers reviewed the titles and abstracts of the identified articles, excluding those that did not meet the predefined eligibility criteria. Articles deemed potentially eligible underwent a subsequent full-text analysis by the same four reviewers to ensure they satisfied the inclusion criteria.

Our systematic search across multiple databases yielded a total of 262 publications: PUBMED (136 articles), OVID (122 articles), and Web of Science (4 articles). After removing 110 duplicates, 152 unique articles remained for screening. Of these, 131 articles were excluded based on the exclusion criteria: non-*EVC* gene reporting (25 articles), no genetic testing performed (64 articles), non-human studies (22 articles), lack of clinical data (17 articles), non-Ellis-van Creveld cases (1 article), and no genetic–corresponding phenotype (2 articles). Ultimately, 15 articles met the inclusion criteria and were included in the final review (Figure 2).

### 3.4. Patient Phenotypes

The objective of this systematic review is to identify potential genotype–phenotype correlations in *EVC*. After applying exclusion criteria, phenotypic data from 66 patients across 49 distinct families were analyzed and summarized. Of the included patients, 35 were male (53.03%), 27 were female (40.9%), and 4 had an unspecified sex (6.06%) due to preterm death. Consanguinity was reported in 19 families (38.77%), absent in 29 families (59.18%), and not documented in 1 family (2.04%). The most common clinical manifestations included postaxial polydactyly ($n = 63$, 95.5%), nail hypoplasia ($n = 45$, 68.2%), congenital heart disease ($n= 45$, 66.7%), rhizomelic shortening ($n = 41$, 62.1%), short ribs with narrow chest ($n = 30$, 45.5%), short frenula ($n = 25$, 37.9%), brachydactyly ($n = 22$, 33.3%), short tubular bones ($n = 21$, 31.8%), hypodontia ($n = 17$, 25.8%), small iliac bones with downward spike ($n = 16$, 24.2%), thin upper lip ($n = 13$, 19.7%), syndactyly ($n = 12$, 18.2%), short broad nose ($n = 12$, 18.2%), fusion of the capitate and hamate ($n = 9$, 13.6%), metaphyseal widening ($n = 9$, 13.6%), serrated alveolar ridge ($n = 8$, 12.1%), coned epiphysis phalanx ($n = 5$, 7.6%), neonatal teeth ($n = 5$, 7.6%), absence of the upper muccobuccal ridge ($n = 5$, 7.6%), axial polydactyly ($n = 3$, 4.5%), bifid tip of the tongue ($n = 3$, 4.5%), fusion of the right proximal tibia and fibula ($n = 2$, 3.0%), central incisor ($n = 2$, 3.0%), bowed humeri ($n = 1$, 1.5%), prominent styloid processes of ulnae ($n = 1$, 1.5%), and scoliosis ($n = 1$, 1.5%). (Figure 3). For most patients, clinical symptoms are documented only when present, but it is often not explicitly stated if a specific clinical characteristic is absent. As a result, these cases are categorized as "not reported" for the corresponding characteristics.

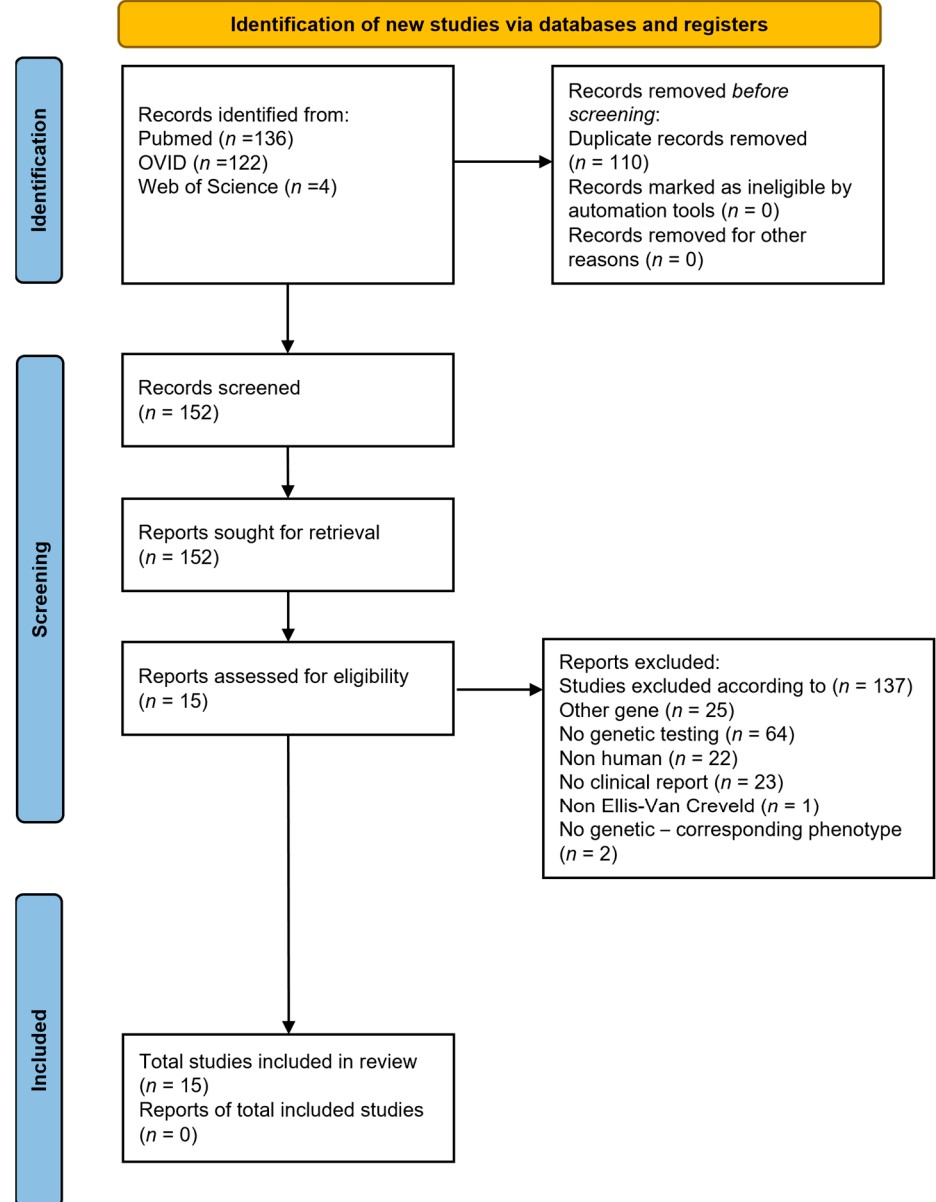

**Figure 2.** PRISMA flowchart illustrating the selection process of reports included in this systematic review. Literature searches were conducted across research databases from July 2024 to December 2024. The screening process was carried out independently by four researchers using Rayyan® software.

### 3.5. Cardiovascular Phenotypes

Congenital heart disease is the third most common clinical feature of EvC syndrome, affecting 66.7% of patients, yet it remains the leading cause of premature death in this population. In our analysis, the most prevalent CHD was atrioventricular canal ($n = 12$, 18.18%), which was followed by atrial septal defects ($n = 10$, 15.15%), common atrium ($n = 7$, 10.61%), functionally univentricular heart ($n = 6$, 9.09%), mitral valve defects ($n = 2$, 3.03%), aortic hypoplasia ($n = 1$, 1.52%) and ventricular septal defects ($n = 1$, 1.52%) (Figure 4). Due to limited detail in the original clinical reports, cardiac subclassifications were not consistently included in this review. Notably, atrioventricular canal and functionally univentricular heart—which collectively account for 28.78% of cases ($n = 19$)—are classified as complex congenital heart diseases, which are associated with high morbidity and mortality. Additionally, eleven different cardiovascular anomalies—occurring as part of the primary congenital heart malformations—were reported in 10 patients (15.15%), including

aortic hypoplasia (*n* = 3; 4.54%), aortic coarctation (*n* = 2; 3.03%), patent ductus arteriosus (*n* = 2; 3.03%), interrupted aortic arch (*n* = 1; 1.51%), double superior vena cava (*n* = 1; 1.51%), persistent left superior vena cava (*n* = 1; 1.51%), and pulmonary hypoplasia (*n* = 1; 1.51%).

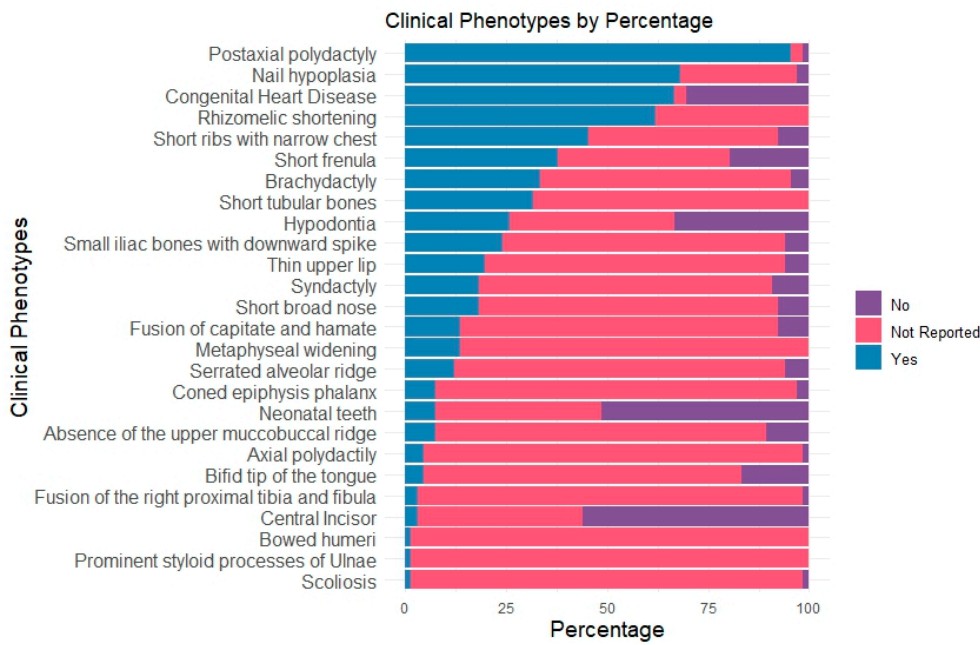

**Figure 3.** Clinical phenotypes ranked by prevalence based on the literature review, highlighting the four most frequently reported features in Ellis-van Creveld syndrome (prevalence > 50%): postaxial polydactyly, nail hypoplasia, congenital heart disease, and rhizomelic shortening.

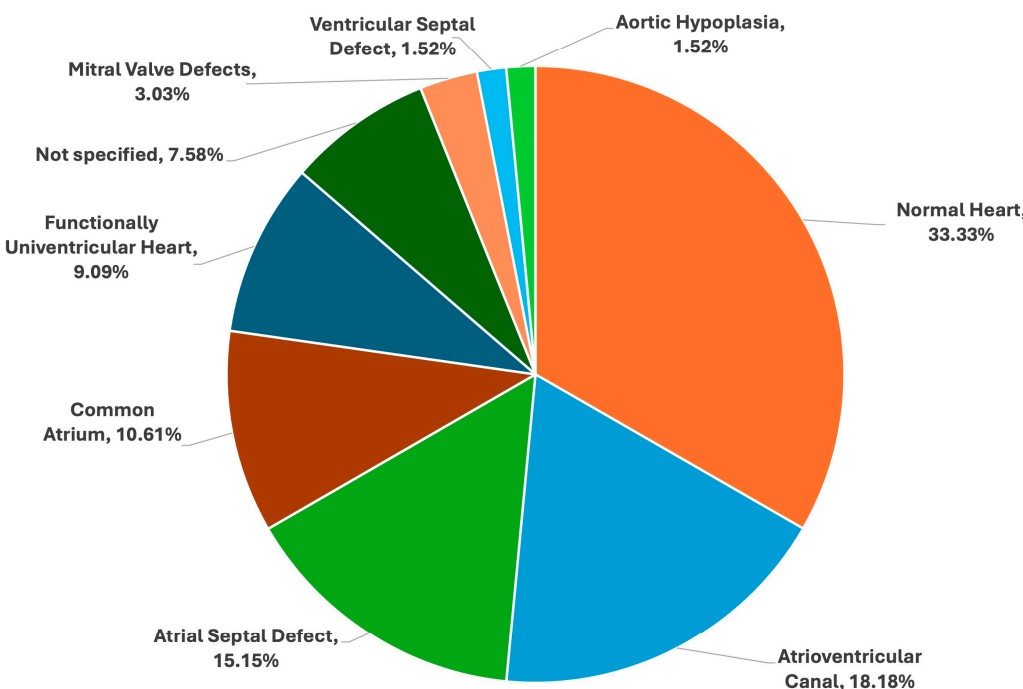

**Figure 4.** Distribution of cardiac phenotypes based on the literature review, highlighting atrioventricular canal as the most frequent congenital heart disease, which was followed by atrial septal defect and common atrium.

### 3.6. Variant Analysis

A total of 53 distinct variants across 132 alleles were analyzed in 66 patients. No specific hotspot was identified, as the variants are distributed throughout the entire gene with the exception of exons 18 and 19, where no variants have been reported (Figure 5). The most frequent variant observed was *EVC* (NM_153717.3):c.1678G>T, which was present in 10 alleles (7.58%). When categorized by variant type, deletions were the most common (*n* = 33, 25.00%), which were followed by nonsense variants (*n* = 30, 22.73%), intronic variants (*n* = 24, 18.18%), missense variants (*n* = 24, 18.18%), duplications (*n* = 12, 9.09%), CNVs (*n* = 8, 6.06%) and insertions (*n* = 1, 0.76%).

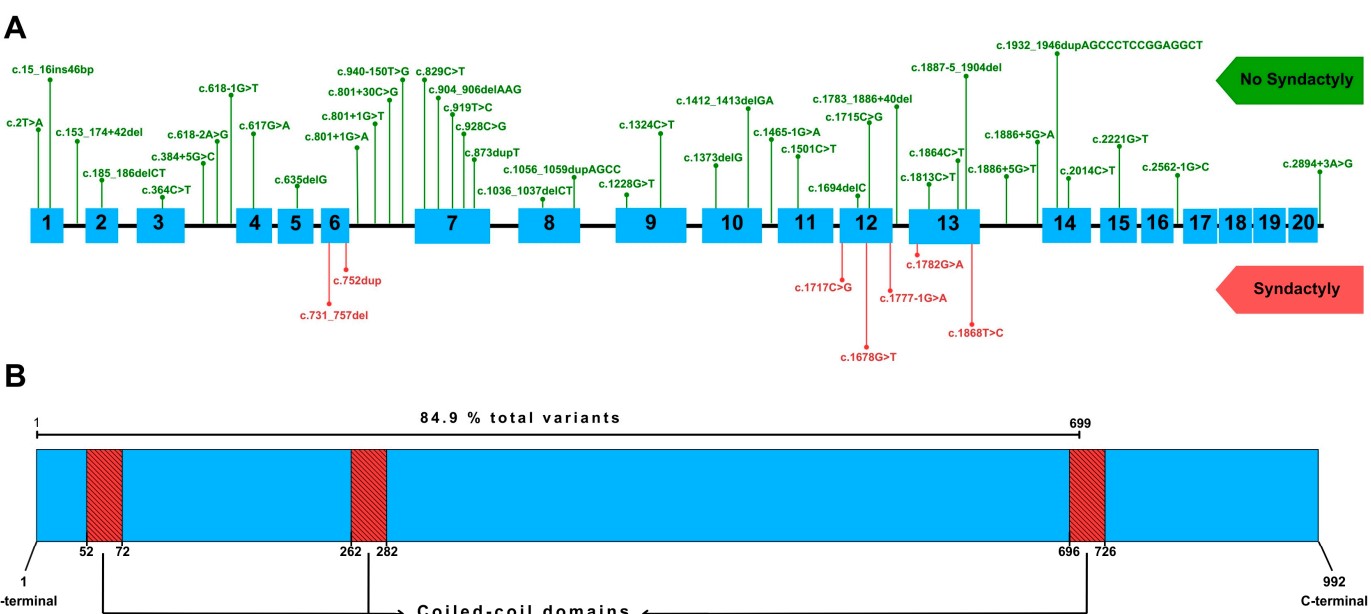

**Figure 5.** (**A**) Schematic representation of the *EVC* gene, mapping the variants analyzed in this systematic review. Variants associated with syndactyly are highlighted in red and are in exons 6, 12, and 13. (**B**) Schematic representation of the EVC protein, illustrating the positions of the three coiled-coil domains. Most variants (84.5%) are clustered within the first 699 amino acids in the N-terminal region. Additionally, pathogenic variants correlated with syndactyly are shown in proximity to the second and third coiled-coil domains.

### 3.7. Variant Position vs. Phenotype

To investigate the potential relationship between genotype and phenotype in EvCS, we systematically mapped all reported clinical manifestations and their corresponding genetic variants onto both the gene and protein structures. This integrative approach aimed to identify potential variant hotspots across the gene and the protein, determine whether specific regions of the protein are associated with particular malformations characteristic of EvCS, and investigate potential correlations between the functional domains of the protein and the location of pathogenic variants (Figure 5).

Variants are broadly distributed across the gene, except for exons 18 and 19, where none have been reported. Notably, the majority cluster in the N-terminal region of the protein, affecting the extracellular domain (amino acids 1–25), the transmembrane domain (amino acids 26–48), and a substantial portion of the cytoplasmic domain (amino acids 49–992). In contrast, fewer pathogenic variants have been identified in the C-terminal region. A possible explanation is that the EVC protein forms a dimeric coiled-coil complex with EVC2, and variants near the N-terminal region may disrupt the three-dimensional structure of this complex. However, this hypothesis has not yet been confirmed by functional studies. Some potential hotspots for syndactyly have been identified in exons 6, 12, and

13. However, no specific protein function has been attributed to the corresponding amino acid regions. Further studies are necessary to investigate this correlation and determine its biological significance.

## 4. Discussion

This systematic review includes 65 previously reported patients with *EVC* variants and phenotypic data along with a newly identified patient carrying a non-previously reported variant. Clinical data and variant information were systematically collected and filtered based on protocol criteria. Variants were then mapped onto the *EVC* gene sequence, revealing their distribution throughout the entire gene, which supports the representativeness of our study.

EVC and EVC2 form a transmembrane heterodimer that regulates Hedgehog signaling from primary cilia. EVC contains a predicted signal anchor sequence, whereas EVC2 has a predicted signal peptide sequence. Both proteins share a key structural feature: C-terminal coiled-coil regions located beyond their transmembrane domains. In EVC, three coiled-coil domains are present at amino acid positions 52–72, 262–282, and 696–724. In this study, 55 distinct variants in *EVC* were identified, with the majority (84.9%) clustering within the first 699 amino acids of the N-terminal region, encompassing all three coiled-coil domains. In contrast, fewer variants were found in the C-terminal region, where no coiled-coil domains are present. This distribution suggests that N-terminal variants may disrupt EVC/EVC2 heterodimer formation, potentially contributing to the Ellis-van Creveld phenotype. This hypothesis is supported by previous proteomic evidence, which demonstrated that deletion of the N-terminal sequence abolishes the interaction between EVC and EVC2 [9].

The EVC/EVC2 heterodimer plays an important role in embryonic development, being expressed from early stages, including the epiblast, gastrulation, neural plate, and all three germ layers. Cilia and the Hedgehog signaling pathway are essential for forelimb and hindlimb development, neural epithelial cells, dental epithelium, facial bones, and chondroblasts [12], which explains the skeletal and dental manifestations of Ellis-van Creveld syndrome. However, the role of EVC/EVC2 in CHD remains poorly understood, despite its presence in 66.7% of patients, representing a major cause of morbidity and mortality. Ciliary genes and hedgehog pathways have been implicated in the pathogenesis of CHD [32], with strong associations observed in atrioventricular canal defects [33], and other syndromic ciliopathies [34]. SHH signaling has been linked specifically to endocytic trafficking and cell signaling processes that influence left–right axis determination during embryogenesis, which is a critical step in proper cardiac morphogenesis [35].

Nonetheless, the specific contribution of the EVC/EVC2 heterodimer to complex CHD remains unclear, particularly in cases associated with poor prognosis. Further research is needed to elucidate its role in cardiac morphogenesis and laterality defects in Ellis-van Creveld syndrome.

Ellis-van Creveld syndrome is a chondroectodermal dysplasia classified under the category of «Skeletal disorders caused by abnormalities of cilia or ciliary signaling» in the latest nosology of genetic skeletal disorders [3]. This category includes 61 distinct diagnoses, which are all linked to genes involved in ciliopathies. EvC syndrome has been associated with at least six different genes (*EVC*, *EVC2*, *WDR35*, *DYNC2LI1*, *GLI1*, and *SMO*). Additionally, there is ongoing debate about whether certain EvC-like phenotypes should be classified within this syndrome or as distinct genetic disorders. As a chondroectodermal dysplasia, EvC syndrome is characterized by skeletal, orofacial, cardiac, and nail abnormalities. The most frequently reported skeletal features in this study include postaxial polydactyly (95.5%), rhizomelic shortening (62.1%), short ribs with a narrow chest

(45.5%), brachydactyly (33.3%), and syndactyly (18.2%). These findings are consistent with previously published data [16]. However, an important limitation in the literature is the lack of detailed radiological descriptions, as key radiographic hallmarks of EvC syndrome are frequently absent from case reports. Notably, findings such as small iliac bones with a downward spike (24.2%), capitate–hamate fusion (13.6%), metaphyseal widening (13.6%), coned epiphyses of the phalanges (7.6%), proximal tibia–fibula fusion (3.0%), and prominent styloid processes of the ulnae (1.5%) are underreported, despite being important for the accurate diagnosis and characterization of the syndrome.

Ectodermal dysplasia presents a broad clinical spectrum, encompassing orofacial features and nail dysplasia, both of which are key phenotypic characteristics of Ellis-van Creveld (EvC) syndrome. In this study, the most frequent ectodermal manifestations observed were nail hypoplasia (68.2%), short frenula (37.9%), hypodontia (25.8%), thin upper lip (19.7%), short broad nose (18.2%), serrated alveolar ridge (12.1%), neonatal teeth (7.6%), absence of the upper mucobuccal ridge (7.6%), bifid tip of the tongue (4.5%), and central incisor anomaly (3.0%). Although previous case reports and cohort studies describe a higher frequency of orofacial abnormalities, our systematic review found that less than 40% of patients exhibited these features. This discrepancy may reflect underreporting or variability in clinical assessment across studies. However, nail dysplasia remains the second most frequently reported clinical feature of EvC syndrome in the literature, which aligns with findings from previously published clinical cohorts [16,29].

In this review, 46 out of 66 EvC syndrome patients presented with congenital heart disease (CHD), and it remains one of the leading causes of mortality, especially in those patients that require cardiac surgery [36]. The most frequently reported CHD in the literature is atrioventricular canal (18.18%), which is followed by atrial septal defects (15.15%) and common atrium (10.61%). These findings align with previously published cohorts that include EvC syndrome patients with pathogenic variants in either *EVC* or *EVC2* [16,36,37]. Our study identifies a higher prevalence of functionally univentricular heart (FUH) (9.09%) in EvC syndrome patients carrying pathogenic variants exclusively in the *EVC* gene. However, the clinical reports analyzed lack a detailed subclassification of the type of CHD underlying the FUH, making it difficult to determine whether a specific subtype is consistently associated with this genetic background.

In the genotype–phenotype correlation analysis, all clinical characteristics were mapped to their corresponding *EVC* variant positions within the gene and protein. No significant associations were found for key clinical features, including nail hypoplasia, postaxial polydactyly, and congenital heart disease. However, a potential correlation was identified between syndactyly and variants located in exons 6, 12, and 13 (Figure 5B). Exon 6 of *EVC* encodes amino acids 235–267, which is a region that includes part of the second coiled-coil domain of the EVC protein (262–282 amino acids). Meanwhile, exons 12 and 13 encode amino acids without a currently defined domain, but they are proximal to the third coiled-coil domain (696–726 amino acids). Although this finding suggests a potential genotype–phenotype correlation, it should be interpreted with caution, as syndactyly appears to be underreported (18.2%) in the literature. The Hedgehog signaling pathway is a key regulator of limb development and has been implicated in syndactyly formation [38]. However, the precise role of EVC within this pathway and its potential contribution to syndactyly in Ellis-van Creveld syndrome remains unclear. Further functional studies and larger patient cohorts are necessary to validate this association and to better understand the potential role of these variants in the pathogenesis of syndactyly in Ellis-van Creveld syndrome.

To our knowledge, this is the third reported Mexican family with Ellis-van Creveld syndrome caused by biallelic pathogenic variants in *EVC* and the first with a Mayan back-

ground. Previous reports [25] described two families and five patients with *EVC*-related EvC syndrome. All cases share key clinical features, including polydactyly, nail hypoplasia, and a narrow chest. However, the cardiological phenotype differs significantly. While the previously reported cases exhibited atrial septal defects in two patients, our case presents a more complex cardiac anomaly, including a single atrium, a single atrioventricular valve with insufficiency, and a ventricular septal defect. These findings highlight the phenotypic variability associated with *EVC* pathogenic variants, particularly in cardiac manifestations, emphasizing the need for further studies to understand potential genotype–phenotype correlations in diverse populations.

Limitations of this systematic review arise from the considerable heterogeneity in the reporting of clinical phenotypes across included studies, which may introduce inconsistencies in data interpretation. Additionally, this study may be prone to publication bias, as it only includes articles published in English and those providing both clinical and molecular diagnoses. As a result, relevant reports lacking molecular confirmation may have been excluded, potentially limiting the generalizability of our findings. Furthermore, some case reports did not provide detailed radiological or cardiovascular descriptions, hindering a comprehensive evaluation of certain skeletal and cardiovascular features. Finally, functional validation of the reported variants was not conducted, highlighting the need for future experimental studies to confirm genotype–phenotype correlations.

## 5. Conclusions

This systematic review provides a comprehensive analysis of the genotype–phenotype correlations in Ellis-van Creveld syndrome, highlighting the high frequency of congenital heart disease, postaxial polydactyly, and nail dysplasia as key clinical features. The identification of 55 distinct variants in *EVC*, predominantly clustered in the N-terminal region, suggests a potential impact on EVC/EVC2 heterodimer function and Hedgehog signaling. Although no clear genotype–phenotype correlation was established for major clinical features, a possible association between syndactyly and variants in exons 6, 12, and 13 was identified, warranting further investigation. However, significant heterogeneity in clinical reporting and the absence of functional studies limit definitive conclusions. Future research should focus on expanding patient cohorts, integrating detailed radiological assessments, and conducting functional validation studies to enhance our understanding of the molecular mechanisms underlying Ellis-van Creveld syndrome.

**Supplementary Materials:** The following supporting information can be downloaded at https://www.mdpi.com/article/10.3390/cardiogenetics15020011/s1, Table S1: Revision Sistematica Ellis van Creveld—English.xlsx [17–20].

**Author Contributions:** S.R.-C.: Medical management of the case, writing the draft of the manuscript. A.-M.C.-E.: Medical management of the case, literature review, revising, reviewing the manuscript critically. C.-D.F.-R.: Literature review, systematic reviewer, writing the initial draft of the manuscript, revising, revising the manuscript critically. P.S.-S.: Literature review, systematic reviewer, writing the initial draft of the manuscript, revising, revising the manuscript critically. C.-G.N.-S.: Literature review, systematic reviewer, writing the initial draft of the manuscript, revising, revising the manuscript critically. H.-A.L.-M.: Literature review, writing the initial draft of the manuscript, revising, revising the manuscript critically, approving the final draft. A.-K.Z.-S.: Medical management of the case, literature review, revising, revising the manuscript critically. F.-J.C.-G.: Medical management of the case, literature review, systematic reviewer, writing the initial draft of the manuscript, conducted genetic and bioinformatic studies, revising, and approving the final draft, revising the manuscript critically. All authors have read and agreed to the published version of the manuscript.

**Funding:** This research was funded by CONAHCYT—Beca Nacional—Funding Number: 2022-000002-01NACF-00427.

**Institutional Review Board Statement:** The study was conducted in accordance with the Declaration of Helsinki and approved by the Institutional Review Board of Hospital General "Dr. Agustín O'Horán in 13 March 2025, approval number CI-015-1-2025".

**Informed Consent Statement:** Informed consent was obtained from parents of the patient involved in the study.

**Data Availability Statement:** All data generated or analyzed during this study are included in this article. Further enquiries can be directed to the corresponding author.

**Conflicts of Interest:** The authors declare no conflict of interest. The funder had no roles in the design if the study; in the collection, analyses, or interpretation of the data; in the writing of the manuscript; or in the decision to publish the results.

## Abbreviations

The following abbreviations are used in this manuscript:

| | |
|---|---|
| EvC | Ellis-van Creveld |
| CHD | congenital heart disease |
| FUH | functionally univentricular heart |
| HH | Hedgehog signaling |

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
