# Peer review of "Genotype–Phenotype Correlation of EVC Variants in Ellis-Van Creveld Syndrome: A Systematic Review and Case Report"

_cardiogenetics, doi:10.3390/cardiogenetics15020011_

Round 1

Reviewer 1 Report

Comments and Suggestions for Authors

EVC syndrome, like other skeletal ciliopathies, is characterized by a highly variable phenotype and patients with different mutations have overlapping phenotypic characteristics. Improved genotype-phenotype characterization of skeletal ciliopathies is essential to improve the multidisciplinary management and long-term prognosis of these patients. Although the objective of the work is interesting, there are a number of considerations for the Authors listed below:

1) The Authors write that the research was done between July 24 to December 24 but in the filters of the search for papers from which year to which year they were considered for inclusion? Specify it in the text. (e.g. papers published in the last 10 years)

2) The aim was to evaluate the correlation between the location of the variant within the EVC gene and specific clinical manifestations, but in the paragraph dedicated to variants vs phenotype it is not well specified what the result of the analysis conducted is, except what is shown in Figure 5. It would be useful to write in this section in more detail what is written briefly only in the discussion “In the genotype-phenotype correlation analysis, all clinical characteristics were mapped to their corresponding EVC variant positions within the gene and protein. No significant associations were found for key clinical features, including nail hypoplasia, postaxial polydactyly, and congenital heart disease”.

3) What type of AVCD was found most frequently? It is now known that cardiac malformations are diagnosed in about 2/3 of affected patients, prevalently AVCD associated with common atrium and systemic and pulmonary venous abnormalities (Digilio MC et al. Atrioventricular canal defect as a sign of laterality defect in Ellis-van creveld and polydactyly syndromes with ciliary and hedgehog signaling dysfunction. Pediatr Cardiol. 2012;33(5):874– 5. https://doi.org/10.1007/s00246-012-0270-3). In the supplementary table, cases of a common atrium and abnormalities of the mitral and tricuspid valve are described as if it were a common atrium with AVCD instead of an isolated common atrium. This would confirm the cardiac phenotype most frequently associated with this syndrome. Could you verify this? And also the echocardiogram described by the Authors in the case report of this manuscript, is suggested for an AVCD (unique atrioventricular valve) with common atrium and aortic hypoplasia. The Authors should underline the particular cardiac phenotype and have to cite the paper “Piceci-Sparascio et al. Common atrium/atrioventricular canal defect and postaxial polydactyly: A mild clinical subtype of Ellis-van Creveld syndrome caused by hypomorphic mutations in the EVC gene. Hum Mutat. 2020;41(12):2087-2093. doi: 10.1002/humu.24112 in which they reported a subtype of EVC syndrome caused by hypomorphic mutations in the EVC gene in a family with vertical transmission of AVCD, common atrium, and postaxial polydactyly.

4) Which are the vascular malformations? Moreover, the Authors should not consider vascular malformations into the CHD category, but as a separate phenotypic feature, so exclude them from the total count of CHD.

5) In the discussion (lines 333-337) the Authors report: “However, our study identifies a higher prevalence of functionally univentricular heart (FUH) (9.09%) in EvC syndrome patients with pathogenic variants exclusively in the EVC gene. Further research is needed to determine whether FUH is a distinct feature of EvC syndrome patients with biallelic pathogenic variants in EVC”. This sentence is confusing and incorrect. It is necessary to clarify first what type of CHD affected the "functional univentricular" hearts described. Were unbalanced AVCD or other types of CHD (such as Hypoplastic left heart syndrome, Tricuspid atresia, Double inlet ventricle, Pulmonary/aortic atresia, Ebstein anomaly). If they were unbalanced AVCD, they would still fall into the AVCD phenotype so the interesting data to be described would be the specific cardiac subtype, therefore the high presence of unbalanced AVCD. If instead they were not unbalanced AVCD but other types of CHD, then the data to be emphasized would be the type of CHD that underlies the FUH.

6) The Authors underline that the role of EVC/EVC2 in CHD remains poorly understood but they should also explain that it is well known a link between AVCD and EVC syndrome. A link between AVCD and cilia abnormalities through a specific pathogenetic pathway involving the Hedgehog signaling has been recognized in several syndromes with AVCD. The role of Hedgehog signaling in coordinating multiple aspects of left-right lateralization and cardiovascular growth is well described. In human beings, Sonic Hedgehog pathway dysregulation has a well known impact on different types of AVCD. The Authors should considered citations below:

-Pugnaloni F et al. Genetics of atrioventricular canal defects. Ital J Pediatr. 2020 May 13;46(1):61. doi: 10.1186/s13052-020-00825-4.

-Ferrante MI et al. Oral-facial-digital type I protein is required for primary cilia formation and left-right axis specification. Nat Genet. 2006;38(1):112–7. https://doi.org/10.1038/ng1684.

-Digilio MC et al. Atrioventricular canal defect and genetic syndromes: the unifying role of sonic hedgehog. Clin Genet. 2019;95(2):268–76. https://doi.org/10.1111/cge.13375.

-Digilio MC et al. Atrioventricular canal defect as a sign of laterality defect in Ellis-van creveld and polydactyly syndromes with ciliary and hedgehog signaling dysfunction. Pediatr Cardiol. 2012;33(5):874– 5. https://doi.org/10.1007/s00246-012-0270-3. 

7) Although the work is only on biallelic variants in EVC and EVC2, the primary genes associated with EVC syndrome, it is important to underline in the text, also in the introduction, that cases of EVC with overlapping phenotypes have also been found associated with dominant variants in PRKACA and PRKACB, recessive variants in WRD35, DYNC2LI1, GLI2 and recently, biallelic variants of DYNC2H1. The Authors should consider citations below:

-Palencia-Campos A, et al. Germline and mosaic variants in PRKACA and PRKACB cause a multiple congenital malformation syndrome. Am J Hum Genet. 2020;107:977–88.

-Caparrós-Martín JA, et al. Specific variants in WDR35 cause a distinctive form of Ellis-van Creveld syndrome by disrupting the recruitment of the EvC complex and SMO into the cilium. Hum Mol Genet. 2015;24(14):4126–37.

-Niceta M, et al. Biallelic mutations in DYNC2LI1 are a rare cause of Ellis-van Creveld syndrome. Clin Genet. 2018; 93(3):632–9.

-Palencia-Campos A, et al. GLI1 inactivation is associated with developmental phenotypes overlapping with Ellis van Creveld syndrome. Hum Mol Genet. 2017;26:4556–71. 9.

-Aubert-Mucca M, et al. Ellis-Van Creveld Syndrome: clinical and molecular analysis of 50 individuals. J Med Genet. 2022:2022-108435.

-Piceci-Sparascio F, et al. Clinical variability in DYNC2H1-related skeletal ciliopathies includes Ellis-van Creveld syndrome. Eur J Hum Genet. 2023 Apr;31(4):479-484.

8) The Authors should standardize the supplementary table (eg. in the CHD row sometimes the Authors write yes followed by the CHD and sometimes only the CHD without yes), and there are few typing errors (eg. in the last column row 11 “yesngle atrium” instead of single atrium).

Author Response

Comments 1: The Authors write that the research was done between July 24 to December 24 but in the filters of the search for papers from which year to which year they were considered for inclusion? Specify it in the text. (e.g. papers published in the last 10 years)

Response 1: Thank you for this important observation. In response, we have incorporated the following text:

Only articles published between 2004 and 2024 were considered for inclusion.

Comment: 

2) The aim was to evaluate the correlation between the location of the variant within the EVC gene and specific clinical manifestations, but in the paragraph dedicated to variants vs phenotype it is not well specified what the result of the analysis conducted is, except what is shown in Figure 5.

It would be useful to write in this section in more detail what is written briefly only in the discussion “In the genotype-phenotype correlation analysis, all clinical characteristics were mapped to their corresponding EVC variant positions within the gene and protein. No significant associations were found for key clinical features, including nail hypoplasia, postaxial polydactyly, and congenital heart disease”.

Response: 

We agree that this section needs revision, and we have replaced it with the following, clearer version:

To investigate the potential relationship between genotype and phenotype in EvCS, we systematically mapped all reported clinical manifestations and their corresponding genetic variants onto both the gene and protein structures. This integrative approach aimed to: identify potential variant hotspots across the gene and the protein, determine whether specific regions of the protein are associated with particular malformations characteristic of EvCS, and investigate potential correlations between the functional domains of the protein and the location of pathogenic variants.

Thank you.

Comment: What type of AVCD was found most frequently?

Response: 

Thank you to point out this important issue, however in our revision, clinical description in clinical reports were not exhaustive, and cardiologic diagnosis were pointed as general diagnosis.

We have added this text in the manuscript:

Due to limited detail in the original clinical reports, cardiac subclassifications were not consistently included in this review.

Comment: 

It is now known that cardiac malformations are diagnosed in about 2/3 of affected patients, prevalently AVCD associated with common atrium and systemic and pulmonary venous abnormalities (Digilio MC et al. Atrioventricular canal defect as a sign of laterality defect in Ellis-van creveld and polydactyly syndromes with ciliary and hedgehog signaling dysfunction.

Pediatr Cardiol. 2012;33(5):874– 5. https://doi.org/10.1007/s00246-012-0270-3).

In the supplementary table, cases of a common atrium and abnormalities of the mitral and tricuspid valve are described as if it were a common atrium with AVCD instead of an isolated common atrium. This would confirm the cardiac phenotype most frequently associated with this syndrome. Could you verify this?

Response: 

We thank the reviewer for this valuable observation.

In both the supplementary table and the case reports analyzed in this review, “atrioventricular canal” and “common atrium” were reported as distinct diagnoses. To avoid confusion, we have updated the supplementary table, clearly separating cardiac malformations and vascular anomalies, our classification and cardiovascular diagnosis criteria is based on the International Pediatric and Congenital Cardiac Code (IPCCC), citation of this classification has been added to the article.

“Clinical phenotype data were categorized into the following variables: congenital heart disease (based on the International Pediatric and Congenital Cardiac Code; IPCCC [31]),”

  1. Béland, M.J.; Franklin, R.C.G.; Aiello, V.D.; Houyel, L.; Weinberg, P.M.; Anderson, R.H. Nomenclature and Classification of Cardiac Defects. In Pediatric and Congenital Cardiology, Cardiac Surgery and Intensive Care; Springer London: London, 2020; pp. 1–23.

Additionally, we have incorporated the suggested citation. (citation 34)

Comment: And also the echocardiogram described by the Authors in the case report of this manuscript, is suggested for an AVCD (unique atrioventricular valve) with common atrium and aortic hypoplasia. The Authors should underline the particular cardiac phenotype and have to cite the paper “Piceci-Sparascio et al. Common atrium/atrioventricular canal defect and postaxial polydactyly: A mild clinical subtype of Ellis-van Creveld syndrome caused by hypomorphic mutations in the EVC gene. Hum Mutat. 2020;41(12):2087-2093. doi: 10.1002/humu.24112 in which they reported a subtype of EVC syndrome caused by hypomorphic mutations in the EVC gene in a family with vertical transmission of AVCD, common atrium, and postaxial polydactyly.

Response: Thank you for this important information.

After reanalysis of the cardiovascular phenotype of the clinical case, both pediatric cardiologist that co-author this manuscript, consider that there is a lack of an important point in the cardiovascular phenotype, we have added “anterior malalignment ventricular septal defect” in the clinical description to clearly differentiate the diagnosis as “Common atrium”.

Also we have added the proposed citation. (citation No. 8)

Comment: Which are the vascular malformations? Moreover, the Authors should not consider vascular malformations into the CHD category, but as a separate phenotypic feature, so exclude them from the total count of CHD.

Response: Thank you for this valuable and important observation.

We have updated the supplementary table to clearly distinguish between cardiac malformations and vascular anomalies. Additionally, we conducted a thorough reanalysis of the cardiac malformations to ensure that both the reported frequencies and proportions remain consistent and accurate.

To reflect these changes, we have also revised the section previously titled "Cardiac Phenotypes", now referred to as "Cardiovascular Phenotypes", and added the frequencies of vascular anomalies. The following text has been incorporated into the revised section:

“In addition, eleven different vascular anomalies were reported in 10 patients (15.15%), including: Aortic hypoplasia (n = 3; 4.54%), aortic coarctation (n = 2; 3.03%), patent ductus arteriosus (n = 2; 3.03%), interrupted aortic arch (n = 1; 1.51%), double superior vena cava (n = 1; 1.51%), left superior vena cava (n = 1; 1.51%), and pulmonary hypoplasia (n = 1; 1.51%).”

Comment: In the discussion (lines 333-337) the Authors report: “However, our study identifies a higher prevalence of functionally univentricular heart (FUH) (9.09%) in EvC syndrome patients with pathogenic variants exclusively in the EVC gene. Further research is needed to determine whether FUH is a distinct feature of EvC syndrome patients with biallelic pathogenic variants in EVC”.

This sentence is confusing and incorrect. It is necessary to clarify first what type of CHD affected the "functional univentricular" hearts described. Were unbalanced AVCD or other types of CHD (such as Hypoplastic left heart syndrome, Tricuspid atresia, Double inlet ventricle, Pulmonary/aortic atresia, Ebstein anomaly). If they were unbalanced AVCD, they would still fall into the AVCD phenotype so the interesting data to be described would be the specific cardiac subtype, therefore the high presence of unbalanced AVCD. If instead they were not unbalanced AVCD but other types of CHD, then the data to be emphasized would be the type of CHD that underlies the FUH.

Response: After reviewing this comment, we agree that the original wording in this section of the discussion may be misleading. Rather than removing it entirely, we have revised the paragraph to improve clarity and ensure the message is accurately conveyed.

“Our study identifies a higher prevalence of functionally univentricular heart (FUH) (9.09%) in EvC syndrome patients carrying pathogenic variants exclusively in the EVC gene. However, clinical reports analyzed lack detailed subclassification of FUH, making it difficult to determine whether an specific subtype is consistently associated with this genetic background.”

Thank you.

Comment. 

The Authors underline that the role of EVC/EVC2 in CHD remains poorly understood but they should also explain that it is well known a link between AVCD and EVC syndrome. A link between AVCD and cilia abnormalities through a specific pathogenetic pathway involving the Hedgehog signaling has been recognized in several syndromes with AVCD. The role of Hedgehog signaling in coordinating multiple aspects of left-right lateralization and cardiovascular growth is well described. In human beings, Sonic Hedgehog pathway dysregulation has a well known impact on different types of AVCD. The Authors should considered citations below:

-Pugnaloni F et al. Genetics of atrioventricular canal defects. Ital J Pediatr. 2020 May 13;46(1):61. doi: 10.1186/s13052-020-00825-4.

-Ferrante MI et al. Oral-facial-digital type I protein is required for primary cilia formation and left-right axis specification. Nat Genet. 2006;38(1):112–7. https://doi.org/10.1038/ng1684.

-Digilio MC et al. Atrioventricular canal defect and genetic syndromes: the unifying role of sonic hedgehog. Clin Genet. 2019;95(2):268–76. https://doi.org/10.1111/cge.13375.

-Digilio MC et al. Atrioventricular canal defect as a sign of laterality defect in Ellis-van creveld and polydactyly syndromes with ciliary and hedgehog signaling dysfunction. Pediatr Cardiol. 2012;33(5):874– 5. https://doi.org/10.1007/s00246-012-0270-3.

Response: 

Thank you for this important comment. We have revised and rewritten this section of the discussion, incorporating the suggested citations.

However, the role of EVC/EVC2 in CHD remains poorly understood, despite its presence in 66.7% of patients, representing a major cause of morbidity and mortality. Ciliary genes and hedgehog pathways have been implicated in the pathogenesis of CHD [31], with strong associations observed in atrioventricular canal defects [32], and other syndromic ciliopathies [33]. SHH signaling,  has been linked specifically to endocytic trafficking and cell signaling processes that influence left-right axis determination during embryogenesis, a critical step in proper cardiac morphogenesis [34].

Comment. 

Although the work is only on biallelic variants in EVC and EVC2, the primary genes associated with EVC syndrome, it is important to underline in the text, also in the introduction, that cases of EVC with overlapping phenotypes have also been found associated with dominant variants in PRKACA and PRKACB, recessive variants in WRD35, DYNC2LI1, GLI2 and recently, biallelic variants of DYNC2H1. The Authors should consider citations below:

-Palencia-Campos A, et al. Germline and mosaic variants in PRKACA and PRKACB cause a multiple congenital malformation syndrome. Am J Hum Genet. 2020;107:977–88.

-Caparrós-Martín JA, et al. Specific variants in WDR35 cause a distinctive form of Ellis-van Creveld syndrome by disrupting the recruitment of the EvC complex and SMO into the cilium. Hum Mol Genet. 2015;24(14):4126–37.

-Niceta M, et al. Biallelic mutations in DYNC2LI1 are a rare cause of Ellis-van Creveld syndrome. Clin Genet. 2018; 93(3):632–9.

-Palencia-Campos A, et al. GLI1 inactivation is associated with developmental phenotypes overlapping with Ellis van Creveld syndrome. Hum Mol Genet. 2017;26:4556–71. 9.

-Aubert-Mucca M, et al. Ellis-Van Creveld Syndrome: clinical and molecular analysis of 50 individuals. J Med Genet. 2022:2022-108435.

-Piceci-Sparascio F, et al. Clinical variability in DYNC2H1-related skeletal ciliopathies includes Ellis-van Creveld syndrome. Eur J Hum Genet. 2023 Apr;31(4):479-484.

Response. 

Thank you for your valuable comment. We have incorporated all the suggested changes and references into the introduction section.

“EvC syndrome is classified as Skeletal disorders caused by abnormalities of cilia or ciliary signaling in the latest nosology of genetic skeletal disorders [3]. This category includes 61 distinct diagnoses, six of which correspond to chondroectodermal dysplasia or EvC syndrome. The syndrome is primarily caused by biallelic pathogenic variants in EVC or EVC2. Additionally, other genes such as WDR35, DYNC2LI1, GLI1, and SMOhave been implicated in related ciliopathies [4–6]. Several conditions previously associated with EvC syndrome are now classified under short-rib thoracic dysplasia, a group of skeletal dysplasias with overlapping clinical features [3]. Moreover, an increasing number of EvC-like phenotypes have been linked to heterozygous pathogenic variants in PRKACA and PRKACB [7], and homozygous pathogenic variants in DYNC2H1 [8].”

Comment. The Authors should standardize the supplementary table (eg. in the CHD row sometimes the Authors write yes followed by the CHD and sometimes only the CHD without yes), and there are few typing errors (eg. in the last column row 11 “yesngle atrium” instead of single atrium).

Response. Thank you for this observation. We have updated the supplementary table to incorporate all suggestions provided by both reviewers.

Reviewer 2 Report

Comments and Suggestions for Authors

Systematic Review

Genotype – Phenotype Correlation of EVC Variants in Ellis-van Creveld Syndrome: A Systematic Review and Case Report

Authors present a case report and PRISMA-based systematic review of the published literature summarizing the genotype-phenotype correlations in EVC (MIM 604831) related Ellis-van Creveld syndrome.

Overall, a good summary listing all currently reported EVC variants in EvC, and the associated range of presenting symptoms.  However, case report is lacking information typically needed (see comments below).

Comment/Suggestions

Line 16: "(EvC) is a rare genetic disorder (7:10,000,000)"

Line 36: " estimated prevalence of approximately 7 cases per 10 million individuals. "

Comment: Citation needed; literature would suggest a much more common prevalence.  e.g. the incidence of Ellis-van Creveld syndrome is estimated at 1 in 60,000 (https://omim.org/entry/225500)

Line 41: "The syndrome is primarily caused by biallelic pathogenic variants in EVC or EVC2, with additional genes (WDR35, DYNC2LI1, GLI1, and SMO) implicated in related ciliopathies."

Comment: citation(s) needed

Line 42: "Some disorders previously linked to EvC syndrome are now classified under short-rib thoracic dysplasia, a group of skeletal dysplasias with overlapping clinical features. "

Comment: citation(s) needed

Page 2 First 4 Paragraphs:

Comment: 4 paragraphs, only 3 citations total between them.  Better/more citing is expected in a review article.

Line 77: "We present the case of a female newborn from Southeast Mexico with a novel mis- sense variant in the EVC gene."

Comment: This information is missing from the abstract.

Line 116: Supplementary excel sheet:

Comment: suggest providing better citation for Authors.  Instead of "last name, year" include PMID or other better article identifier. 

Comment: suggest change "Variant Described" to "EVC Variant Described".  Were all Variants EVC? if not need to differentiate between EVC and EVC2 genes.

Line 119: "2.5. Outcome variables"

Comment: study limitations should be left for discussion.

Line 129: "3.1. Case Description:"

Comment: A case report should contain more than the simple immediate findings.  Missing is:

  • Medical, family, and psycho-social history including relevant genetic information (of parents?)
  • Types of therapeutic intervention (such as pharmacologic, surgical, preventive)
  • Administration of therapeutic intervention (such as dosage, strength, duration)
  • Clinician and patient-assessed outcomes (if available)
  • Important follow-up diagnostic and other test results
  • Adverse and unanticipated events

Line 157: "Case Description: Genetic Analysis "

Comment: being methods, this should be moved to methods section (~Line 82)

Line 231: Figure 4.

Comment: suggest adding the %values into legend labels (or adding labels directly on pie graph).  Colours could be confusing for some readers.

Line 258: Figure 5.

Comment: green text in Fig 5a is a little small/pixelated.  Not enough to change but would suggest providing higher resolution image of this as a supplemental figure (or perhaps a supplemental table listing all variants).

Author Response

Comment. 

Line 16: "(EvC) is a rare genetic disorder (7:10,000,000)"

Line 36: " estimated prevalence of approximately 7 cases per 10 million individuals. "

 Comment: Citation needed; literature would suggest a much more common prevalence.  e.g. the incidence of Ellis-van Creveld syndrome is estimated at 1 in 60,000 (https://omim.org/entry/225500)

Response: 

Thank you for this important observation.

Incidence reports for EvC are scarce. We agree that there is an incidence report from D’Asdia, 2012 from a highly consanguinity populations, which has been added to the manuscript:

“Ellis-van Creveld (EvC) syndrome (OMIM: 225500) is a rare chondroectodermal dysplasia with an estimated prevalence of approximately 7 cases per 10 million individuals. However, epidemiological studies in specific populations report significantly higher prevalence rates, reaching up to 5 cases per 1,000 live births [1], and as high as 1 in 60,000 live births in populations with a elevated rates of consanguinity [2].”

We have added the next citation, supporting the incidence:

D'Asdia MC, Torrente I, Consoli F, Ferese R, Magliozzi M, Bernardini L, Guida V, Digilio MC, Marino B, Dallapiccola B, De Luca A. Novel and recurrent EVC and EVC2 mutations in Ellis-van Creveld syndrome and Weyers acrofacial dyostosis. Eur J Med Genet. 2013 Feb;56(2):80-7. doi: 10.1016/j.ejmg.2012.11.005. Epub 2012 Dec 7. PMID: 23220543. /// le agregamos la cita de donde saca la info la autora que sugirió la revisora: Galdzicka M, Patnala S, Hirshman MG, Cai JF, Nitowsky H, Egeland JA, Ginns EI. A new gene, EVC2, is mutated in Ellis-van Creveld syndrome. Mol Genet Metab. 2002 Dec;77(4):291-5. doi: 10.1016/s1096-7192(02)00178-6. PMID: 12468274.

Comment: 

Line 41: "The syndrome is primarily caused by biallelic pathogenic variants in EVC or EVC2, with additional genes (WDR35, DYNC2LI1, GLI1, and SMO) implicated in related ciliopathies."

 Comment: citation(s) needed

Response: 

Thank you for this important observation. As this comment aligns with a point previously raised by Reviewer No. 1, we have revised the corresponding section accordingly and incorporated several relevant citations to strengthen the introduction.

“EvC syndrome is classified as Skeletal disorders caused by abnormalities of cilia or ciliary signaling in the latest nosology of genetic skeletal disorders [3]. This category includes 61 distinct diagnoses, six of which correspond to chondroectodermal dysplasia or EvC syndrome. The syndrome is primarily caused by biallelic pathogenic variants in EVC or EVC2. Additionally, other genes such as WDR35, DYNC2LI1, GLI1, and SMOhave been implicated in related ciliopathies [4–6]. Several conditions previously associated with EvC syndrome are now classified under short-rib thoracic dysplasia, a group of skeletal dysplasias with overlapping clinical features [3]. Moreover, an increasing number of EvC-like phenotypes have been linked to heterozygous pathogenic variants in PRKACA and PRKACB [7], and homozygous pathogenic variants in DYNC2H1 [8].”

Comment. 

Line 42: "Some disorders previously linked to EvC syndrome are now classified under short-rib thoracic dysplasia, a group of skeletal dysplasias with overlapping clinical features. "

Comment: citation(s) needed

Response. 

This citation has been added to this statement (citation no. 3):

Unger, S.; Ferreira, C.R.; Mortier, G.R.; Ali, H.; Bertola, D.R.; Calder, A.; Cohn, D.H.; Cormier-Daire, V.; Girisha, K.M.; Hall, C.; et al. Nosology of Genetic Skeletal Disorders: 2023 Revision. Am J Med Genet A 2023, 191, 1164–1209, doi:10.1002/ajmg.a.63132.

Thank you,

Comment. Page 2 First 4 Paragraphs:

Comment: 4 paragraphs, only 3 citations total between them.  Better/more citing is expected in a review article.

Response: 

We have added the next citations to the introduction section:

  1. Barbeito, P.; Martin-Morales, R.; Palencia-Campos, A.; Cerrolaza, J.; Rivas-Santos, C.; Gallego-Colastra, L.; Caparros-Martin, J.A.; Martin-Bravo, C.; Martin-Hurtado, A.; Sánchez-Bellver, L.; et al. EVC-EVC2 Complex Stability and Ciliary Targeting Are Regulated by Modification with Ubiquitin and SUMO. Front Cell Dev Biol 2023, 11, doi:10.3389/fcell.2023.1190258.

  1. Thomas, D.C.; Moorthy, J.D.; Prabhakar, V.; Ajayakumar, A.; Pitchumani, P.K. Role of Primary Cilia and Hedgehog Signaling in Craniofacial Features of Ellis–van Creveld Syndrome. Am J Med Genet C Semin Med Genet 2022, 190, 36–46.

  1. Baujat, G.; Le Merrer, M. Ellis-van Creveld Syndrome. Orphanet J Rare Dis 2007, 2.

  1. Muensterer, O.J.; Berdon, W.; McManus, C.; Oestreich, A.; Lachman, R.S.; Cohen, M.M.; Done, S. Ellis-van Creveld Syndrome: Its History. Pediatr Radiol 2013, 43, 1030–1036.

  1. O’Connor, M.J.; Collins, R.T. Ellis-van Creveld Syndrome and Congenital Heart Defects: Presentation of an Additional 32 Cases. Pediatr Cardiol 2012, 33, 491–492.

Thank you,

Comment. 

Line 77: "We present the case of a female newborn from Southeast Mexico with a novel missense variant in the EVC gene."

Comment: This information is missing from the abstract

Response: 

Thank you for this very important observation, we have modified the abstract adding this information:

“We present a case of a female newborn from Southeast Mexico carrying a novel missense variant in EVC,aligned with a systematic review aimed at exploring genotype-phenotype correlations in EVC-related EvC.”

Comment. 

Line 116: Supplementary excel sheet:

Comment: suggest providing better citation for Authors.  Instead of "last name, year" include PMID or other better article identifier. 

Comment: suggest change "Variant Described" to "EVC Variant Described".  Were all Variants EVC? if not need to differentiate between EVC and EVC2 genes

Response. 

We have added a new row to the table indicating the PMID for each referenced report.Additionally, we updated the row title to “EVC variant described”, as all reported variants pertain specifically to the EVC gene.Thank you for your valuable suggestion.

Comment. 

Line 119: "2.5. Outcome variables"

Comment: study limitations should be left for discussion.

Response. 

We have revised the "Outcome Variables" paragraph and moved the limitations to the Discussion section.

2.6       Outcome variables

The primary objectives of this study were to identify the most frequently reported clinical features of Ellis-van Creveld syndrome and to evaluate potential correlations between variant positions within the genomic sequence and specific clinical manifestations. Data analysis was conducted using R software (v.1.2.5042; R Software, Inc), we systematically mapped all reported clinical findings and their corresponding genetic variants onto both the gene and the protein structures.

Revised Discussion paragraph:

Limitations of this systematic review arise from the considerable heterogeneity in the reporting of clinical phenotypes across included studies, which may introduce inconsistencies in data interpretation. Additionally, this study may be prone to publication bias, as it only includes articles published in English and those providing both clinical and molecular diagnoses. As a result, relevant reports lacking molecular confirmation may have been excluded, potentially limiting the generalizability of our findings. Furthermore, some case reports did not provide detailed radiological or cardiovascular descriptions, hindering a comprehensive evaluation of certain skeletal and cardiovascular features. Finally, functional validation of the reported variants was not conducted, highlighting the need for future experimental studies to confirm genotype-phenotype correlations.

Thank you.

Response. 

Line 129: "3.1. Case Description:"

Comment: A case report should contain more than the simple immediate findings.  Missing is:

  • Medical, family, and psycho-social history including relevant genetic information (of parents?)
  • Types of therapeutic intervention (such as pharmacologic, surgical, preventive)
  • Administration of therapeutic intervention (such as dosage, strength, duration)
  • Clinician and patient-assessed outcomes (if available)
  • Important follow-up diagnostic and other test results

Adverse and unanticipated events

Response: 

Thank you for this valuable observation. We have revised the entire case description and incorporated the suggested information.

We present a 15-day-old female newborn of Mayan-Yucatecan ancestry, with no known history of consanguinity. However, both parents are residents of  a Mayan population, suggesting potential endogamy. The parents are healthy with no history of chronic diseases or relevant surgical interventions. The mother’s pregnancy was confirmed in the first trimester, with routine prenatal care showing no complications, prenatal ultrasounds were normal until the third trimester, when possible fetal growth restriction was noted. The proband was delivered via cesarean section at 39 weeks of gestation, with an APGAR score of 8/9 and no signs of respiratory distress. Physical examination revealed a weight of 2960 g and a length 48 cm (-1.2 SD). The propositus had a normocephalic head (head circumference: 34 cm, -0.2 SD), a broad forehead, horizontal palpebral fissures, neonatal teeth, a short neck, and a narrow thorax. Limb examination showed rhizomesomelic shortening, bilateral post-axial polydactyly in both hands, and axial polydactyly in the right foot (duplication of the 4th toe) with syndactyly of 4th and 5th toes (Figures 1A, 1B). Nail dysplasia was also observed. External genitalia were phenotypically female, with no abnormalities.

Short after the delivery the patient was admitted to the Neonatal Intensive Care Unit (NICU) due to respiratory distress, which required continuous positive airway pressure (CPAP) and nutritional support with parenteral nutrition. After 4 days, the respiratory distress ceased, the patient was admitted to the Pediatrics Department, where she received supplemental oxygen therapy and initiated oral feeding. Due to skeletal abnormalities and bone dysplasia, a referral was made to the Genetics Department. Following clinical evaluation, imaging and diagnostic studies were requested, including whole exome sequencing (WES), which confirmed the diagnosis of Ellis-van Creveld syndrome.

X-rays revealed shortened ribs and iliac bones in the cephalocaudal dimension. The ilium displays a downward-directed, hook-like projection at the greater sciatic notch. The ischial and pubic bones were also shortened, along with notable shortening of the humerus, radius and ulna, accompanied by metaphyseal widening (Figure 1C).

The echocardiogram revealed situs solitus with levocardia, pulmonary veins draining into a single atrium, a single atrioventricular valve with insufficiency, two well-differentiated ventricles, and a small anterior malalignment ventricular septal defect (2 mm). Additionally, aortic hypoplasia was observed throughout the tract including emergence, proximal, and distal arch. Diuretic therapy was initiated with spironolactone (0.5 - 1 mg/kg/day), which was discontinued after a few days due to of hyperkalemia and subsequently replaced with hydrochlorothiazide (1.0 mg/kg/day). Patient suffered sudden death at 20 days of life due to complex congenital heart disease.

Comment: 

Line 157: "Case Description: Genetic Analysis "

Comment: being methods, this should be moved to methods section (~Line 82)

Response: 

Thank you for this valuable observation.

We have created a new methods section “2.2 Clinical Case: genetic analysis” and “3.2 Case Description: results and interpretation of Whole-exome sequencing”

Comment: 

Line 231: Figure 4.

Comment: suggest adding the %values into legend labels (or adding labels directly on pie graph).  Colours could be confusing for some readers.

Response: 

We have updated the graph's colors and composition to enhance clarity and understanding.

Thank you.

Comment: 

Line 258: Figure 5.

Comment: green text in Fig 5a is a little small/pixelated.  Not enough to change but would suggest providing higher resolution image of this as a supplemental figure (or perhaps a supplemental table listing all variants).

Response: We have enhanced the image resolution and changed the color for improved readability.

Thank you,

Round 2

Reviewer 1 Report

Comments and Suggestions for Authors

The Authors have modified the manuscript according to the suggestions made. I have only two comments:

1) In the new paragraph of the FUH (line 390-393), the Authors should rephrase the sentence as follows: "However, clinical reports analyzed lack detailed subclassification of the type of CHD underlying the FUH, making it difficult to determine whether an specific cardiac subtype is consistently associated with this genetic background".

2) There was probably an error in the first version regarding the use of the term “vascular anomalies”. In fact, the vascular anomalies now specified by the Authors, i.e. aortic hypoplasia, aortic coarctation, PDA, aortic arch interruption are actually not classifiable as vascular anomalies but as CHD, except for persistent left superior vena cava (the Authors should add the word “persistent”) and double vena cava. So it would be appropriate for the authors to re-modify the supplementary table by correctly relocating only these last two as vascular anomalies, while leaving the remaining ones in the CHD row. Moreover, pulmonary hypoplasia must be specified whether it is pulmonary artery hypoplasia or pulmonary branch hypoplasia, and must be included in the CHD group.

Author Response

Comment 1: In the new paragraph of the FUH (line 390-393), the Authors should rephrase the sentence as follows: "However, clinical reports analyzed lack detailed subclassification of the type of CHD underlying the FUH, making it difficult to determine whether an specific cardiac subtype is consistently associated with this genetic background".

Response: We agree with this suggestion, and the paragraph now reads:

“However, clinical reports analyzed lack detailed subclassification of the type of CHD undelying the FUH, making it difficult to determine whether an specific subtype is consistently associated with this genetic background.”

Thank you.

Comment 2: 

There was probably an error in the first version regarding the use of the term “vascular anomalies”. In fact, the vascular anomalies now specified by the Authors, i.e. aortic hypoplasia, aortic coarctation, PDA, aortic arch interruption are actually not classifiable as vascular anomalies but as CHD, except for persistent left superior vena cava (the Authors should add the word “persistent”) and double vena cava. So it would be appropriate for the authors to re-modify the supplementary table by correctly relocating only these last two as vascular anomalies, while leaving the remaining ones in the CHD row. Moreover, pulmonary hypoplasia must be specified whether it is pulmonary artery hypoplasia or pulmonary branch hypoplasia, and must be included in the CHD group.   Response: Thank you for this important clarification.

We have updated the supplementary table to reflect the proposed changes.

Regarding D’Asadia (Case 4): The original report only mentions “pulmonary hypoplasia,” without further specification. Therefore, we retained the term as reported.

Updated text:  “Congenital heart disease is the third most common clinical feature of EvC syndrome, affecting 66.7% of patients, yet it remains the leading cause of premature death in this population. In our analysis, the most prevalent CHD was atrioventricular canal (n = 12, 18.18%), followed by atrial septal defects (n = 10, 15.15%), common atrium (n = 7, 10.61%), functionally univentricular heart (n = 6, 9.09%), mitral valve defects (n = 2, 3.03%), aortic hypoplasia (n = 1, 1.52%) and ventricular septal defects (n = 1, 1.52%) (Figure 4).”

Figure 4 has been revised to reflect the updated distribution, now including aortic hypoplasia.

We also rephrased the paragraph describing cardiovascular anomalies. Although we considered limiting it to vascular anomalies only, we decided to retain all descriptions, as we believe this information provides important context.

Additionally, eleven different cardiovascular anomalies - occurring as part of the primary congenital heart malformations- were reported in 10 patients (15.15%), including: Aortic hypoplasia (n = 3; 4.54%), aortic coarctation (n = 2; 3.03%), patent ductus arteriosus (n = 2; 3.03%), interrupted aortic arch (n = 1; 1.51%), double superior vena cava (n = 1; 1.51%), persistent left superior vena cava (n = 1; 1.51%), and pulmonary hypoplasia (n = 1; 1.51%).

Reviewer 2 Report

Comments and Suggestions for Authors

Authors have adequately addressed my previous peer-review feedback and have sufficiently updated text. 

Author Response

We sincerely thank the reviewer for their valuable time and thoughtful evaluation of our manuscript.